# Integrating Complex Permittivity Measurements with Histological Analysis for Advanced Tissue Characterization

**DOI:** 10.3390/s25082626

**Published:** 2025-04-21

**Authors:** Sandra Lopez-Prades, Mónica Torrecilla-Vall-llossera, Mercedes Rus, Miriam Cuatrecasas, Joan M. O’Callaghan

**Affiliations:** 1Pathology Department, Hospital Clinic Barcelona, C/Villarroel 170, 08036 Barcelona, Spain; slopezp@recerca.clinic.cat (S.L.-P.); mcuatrec@clinic.cat (M.C.); 2Faculty of Medicine, University of Barcelona, C/Casanova 143, 08036 Barcelona, Spain; 3Institut d’Investigacions Biomèdiques August Pi i Sunyer (IDIBAPS), C/Roselló 149, 08036 Barcelona, Spain; 4ICFO—Institut de Ciències Fotòniques, The Barcelona Institute of Science and Technology, 08860 Castelldefels, Spain; monica.torrecilla@icfo.eu; 5Centro de Investigación Biomédica en Red de Enfermedades Hepáticas y Digestivas (CIBERehd), Av. Monforte de Lemos, 3-5, 28029 Madrid, Spain; 6CommSensLab, Department of Signal Theory and Communications, Universitat Politècnica de Catalunya (UPC), 08034 Barcelona, Spain

**Keywords:** complex permittivity, biological tissues, open-ended coaxial probe, histology

## Abstract

**Highlights:**

**What are the main findings?**

**What is the implication of the main finding?**

**Abstract:**

We developed a measurement setup and protocol reliably relating complex permittivity measurements with tissue characterization and specific histological features. We measured 148 fresh human tissue samples across 14 tissue types at 51 frequencies ranging from 200 MHz to 20 GHz, using an open-ended coaxial slim probe. Tissue samples were collected using a punch biopsy, ensuring that the sampled area encompassed the region where complex permittivity measurements were performed. This approach minimized experimental uncertainty related to potential position-dependent variations in permittivity. Once measured, the samples were then formalin-fixed and paraffin-embedded (FFPE) to obtain histological slides for microscopic analysis of tissue features. We observed that complex permittivity values are strongly associated with key histological features, including fat content, necrosis, and fibrosis. Most tissue samples exhibiting these features could be differentiated from nominal values for that tissue type, even accounting for statistical variability and instrumental uncertainties. These findings demonstrate the potential of incorporating fast in situ complex permittivity for fresh tissue characterization in pathology workflows. Furthermore, our work lays the groundwork for enhancing databases where complex permittivity values are measured under histological control, enabling precise correlations between permittivity values, tissue characterization, and histological features.

## 1. Introduction

The open-ended coaxial probe has become the preferred method for measuring the dielectric properties of biological tissues due to its simplicity, minimal sample handling, and non-destructive nature. It enables both ex vivo and in vivo measurements across a wide frequency spectrum, spanning δ dispersions (between 0.1 and 5 GHz) caused by the dipolar moment of proteins and other large molecules, and γ dispersions (in the GHz range) due to the presence of water and small molecules [1,2].

Open-ended probes comprise a truncated section of a coaxial transmission line, along which electromagnetic fields can propagate. The reflections at the probe tip depend on the permittivity of the tissue sample in contact with the tip. A predecessor of this technique was introduced in [3], where the dielectric in the final segment of a short-circuited coaxial line was replaced by the material under test. This approach has been widely used to characterize dielectric properties of materials [4]. Various techniques have subsequently evolved and have been adapted to the specifics of biological tissues [5]. A significant improvement to these techniques was provided by Gabriel et al. [6], who were able to relate the reflection coefficient at the probe tip with the dielectric properties of the media under test by matching the boundary conditions at the interface between the coaxial line and the semi-infinite media [7].

The advent of open-ended probes has catalyzed extensive research into the broadband characterization of biological tissues. Gabriel et al. [1] conducted a comprehensive literature review and supplemented existing data with new measurements to compile a database [8], which remains a cornerstone among existing databases [9,10]. Despite its importance, Gabriel’s data acquisition lacked histological control, which could introduce errors due to histological heterogeneity, even in normal tissues. Consequently, it remains the need to relate permittivity with histological features. Even today, there is a relatively scant amount of data relating both, mostly concerning histological alterations caused by cancer or by other pathologies [11,12,13,14,15,16]. Moreover, histological analysis is often performed in areas much larger than the probe’s sensing region, and the sample processing can alter the sample’s volume and morphology. This complicates the correlation between dielectric properties and histological characteristics, leading to the exclusion of a significant number of samples [17,18]. The use of punch biopsies with areas similar to the probe’s footprint could facilitate this correlation and spur the development of comprehensive databases on the complex permittivity of human tissues, incorporating the effects of histological alterations and heterogeneity.

In this article, we address this problem by assembling a measurement workstation adapted to be used in a pathology department. This facilitates access to human tissue samples from surplus diagnostic tissues to combine expert histological analysis with measurements of complex permittivity.

## 2. Materials and Methods

### 2.1. Measurement System

The measurement system used in this study has been designed to combine the guidelines provided by La Gioia et al. [19], to ensure optimal measurement results, and the specific conditions encountered in a pathology department. To address both, we built a complex permittivity workstation cart (Figure 1) using a modular system [20] to provide the following essential features: compactness, stable connections between the vector network analyzer (VNA) and the probe, the ability to bring the sample into contact with a fixed probe, and the possibility of adjusting the sample–probe pressure.

The station integrates a VNA (FieldFox N9918A); a PC for data acquisition, instrument control, and measurement validation; a coarse lift platform with a bolted-in precision lab jack (Thorlabs L200/M); a probe stand integrated into the cart’s structure; a 2.2 mm diameter open-ended coaxial probe [21]; an electronic scale to control the probe-to-sample pressure; and a polytetrafluoroethylene (PTFE) sample holder.

One of the key features required in a broadband complex permittivity system at GHz frequencies is the stability of the electrical properties of the connection between the VNA and the probe. Measurements are highly sensitive to phase and amplitude variations caused by uncontrolled motion of coaxial cables. We addressed this issue by securing the relative positions of the VNA and the probe and using a semi-rigid cable, protected by a plastic tube, between the VNA and the connection point of the probe. The probe stand is built into the cart frame, providing additional mechanical stability and further reducing the risk of uncontrolled cable motion. The VNA is securely mounted to the cart, fitted between two symmetrical, 3D-printed, polylactic acid (PLA) pieces. These pieces can be positioned along an aluminum profile within the modular system to ensure a snug fit for the VNA. The semi-rigid coaxial cable runs between the VNA port, and the upper end of a 3.5 mm female-to-female bulkhead adapter fixed to a horizontal plate on the probe stand. The open-ended probe is connected to the lower part of the bulkhead adapter through a 3.5 to 2.4 mm transition. This setup isolates the cable from probe movements and prevents torque from affecting the connection when attaching the probe to the system.

### 2.2. Instrument Settings, Calibration, and Validation

Before commissioning the workstation to the pathology department, extensive characterization procedures were conducted to determine the optimal instrument settings, measurement protocol, and associated uncertainties. We employed the standard calibration procedure, which involves measuring three standard loads, where the probe is open-circuited, short-circuited, and immersed in deionized water [21]. After calibration, we measured a temperature-controlled 0.1 mol/L NaCl solution and compared the results to reference values derived from a Cole–Cole model, whose parameters vary with both temperature and concentration, as described in [22,23]. This comparison confirmed that electronic noise was not the main source of discrepancies between the measured permittivity and the reference values. Consequently, variations in the VNA averaging, output power, and intermediate frequency bandwidth (IFBW) had no significant impact when adjusted from the instrument’s default settings.

We investigated the potential effect of the IFBW on measurement time. While low IFBW values, typically used for low-noise measurements, can lead to longer measurement times, our measurements were primarily limited by the instrument’s firmware execution time. Therefore, increasing the IFBW provided no appreciable benefit. Based on these findings, we set the VNA output power to −6 dBm, with no averaging, and an IFBW of 300 Hz. To keep the frequency sweep time within reasonable bounds, we limited the number of points per sweep to 51 and used a logarithmic frequency sweep from 200 MHz to 20 GHz to cover two full frequency decades.

Once the instrument settings were finalized, we established the maximum acceptable difference between the measured and reference permittivity values. VNA calibration was repeated if the validation measurement of the 0.1 mol/L NaCl solution, performed immediately after calibration, exceeded this maximum threshold. Frequency-dependent validation thresholds were determined for both the real and imaginary parts of the permittivity by performing 41 calibration–validation sequences and recording the differences between the measured permittivity and its reference value. The thresholds were set to the 80th percentile of the absolute difference for the real part and the 90th percentile for the imaginary part (Figure 2).

Once these thresholds had been established, they remained unchanged over 31 months, and proved useful in detecting wear in the short-circuit standard, which could result in defective calibrations or in identifying a lack of instrument stability due to insufficient warm-up time after being powered on. Between one and four VNA calibrations were needed prior to each measurement session.

### 2.3. Sample Acquisition

All samples used in the study were fresh human diagnostic surplus tissues obtained from surgical specimens or autopsies. Tissue samples were obtained from brain, thyroid, lung, spleen, liver, kidney, salivary gland, fat, skeletal and heart muscle, tongue, and pancreas. Measurements were performed within 30 min of the surgical specimen’s extraction and within 12 h from death for tissues obtained from autopsies. The study was approved by the Ethics Committee of the Hospital Clinic of Barcelona (HCB/2023/920) and conducted in accordance with the Helsinki Declaration. All samples were handled anonymously.

### 2.4. Tissue Measurements

Permittivity measurements of fresh human tissue samples were conducted in the Pathology Department of Hospital Clinic of Barcelona, Spain. These measurements took about 1 min per sample, required no tissue processing, and did not damage or alter the properties of the samples. Each sample was placed on the PTFE base, and the probe-to-sample pressure was adjusted using the precision jack until a 1 g reading was achieved on the scale (Figure 3A). The VNA was then used to measure complex permittivity, employing a 51-point logarithmic frequency sweep from 200 MHz to 20 GHz. All facilities and tissues were at room temperature (from 21 to 24 °C).

We performed three consecutive permittivity measurements on every measurement spot using the coaxial probe and recorded their average (Figure 3A). Then, we performed a 5 mm diameter punch biopsy of the tissue sample, centered on the probe’s footprint area (Figure 3B). The tissue punch was then fixed in formalin and paraffin-embedded (FFPE) to create a paraffin block. A microtome (Tissue-Tek AutoSection, Sakura, Japan) was used to obtain a 2 µm histological section from the paraffin block containing the FFPE tissue. This tissue section was placed on a glass slide and stained with Hematoxylin–Eosin (HE) (Figure 3C). We ensured a strong correspondence between the pathological analysis and the permittivity measurements by verifying that the diameter of the punch in the HE-stained tissue slide differed by no more than 0.5 mm from that of the sample prior to FFPE processing.

The HE slides were reviewed by a pathologist (M.C.), using an optical microscope (Olympus BX41, Olympus®, Japan) to identify the tissue type and to evaluate the homogeneity of the measured area within the selected punch area, by identifying features such as fibrous tracts or large vessels, which could affect permittivity measurements and render the sample invalid for our study. All tumoral samples were also discarded. The histological analysis was also utilized to identify samples with fibrosis, necrosis, or fat content, which were then categorized accordingly. All data were annotated in a database to ensure proper correlation between permittivity values and histological features for both normal samples and samples with histological alterations, i.e., fibrosis, necrosis, or fat content.

### 2.5. Detection of Fat, Necrosis, and Fibrosis Through Permittivity Measurements

To evaluate the potential of complex permittivity for detecting the presence of fibrosis, necrosis, or fat content in samples, we compared the complex permittivity values at 51 frequency points for samples of a given tissue type exhibiting these histological features with the average values of samples of the same tissue type without such features. Detection was possible whenever the real or imaginary parts deviated beyond the expected limits, accounting for instrumental or statistical uncertainties. Statistical uncertainty was defined by the standard deviation of permittivity measurements in samples without the histological feature of interest. Instrumental uncertainty at each frequency was determined through 41 calibration–validation sequences (detailed in Section 2.1), where the standard deviation of differences between NaCl solution measurements and theoretical values was calculated. The 28th-highest absolute difference (corresponding to the 68th percentile under a normal distribution) was used as the representative value of instrumental uncertainty to mitigate outlier effects.

### 2.6. Comparison with a Reference Database

For each tissue type, we compared the average permittivity of standard samples with those of the corresponding human tissue types, as published by Gabriel in 1996 [8], which serves as a reference database (RDB). This database provides permittivity measurements for both animal and human tissues across a broader frequency range than that of our study. Although the maximum frequency in both our dataset and the RDB extends to 20 GHz, the minimum frequency in the RDB varies by tissue type.

Additionally, the RDB data often have sparser frequency sampling for some tissues, with fewer data points distributed over wider frequency ranges (Table 1). In contrast, our dataset includes 51 frequency points with a uniform logarithmic distribution spanning 200 MHz to 20 GHz for all human tissue types analyzed.

We compared our data with the RDB to evaluate the consistency of measurements over the shared frequency range of 200 MHz to 20 GHz. We evaluated three features: (1) whether the data for that tissue type was present in the RDB; (2) if the distribution of our measurement frequencies was denser than that used in the RDB for the same tissue type; (3) if our measurements of complex permittivity from 200 MHz to 20 GHz were consistent with the RDB data within the same frequency range.

Agreement between complex permittivity values and their RDB counterparts was classified as follows: Good if at least 80% of the measurement points fell within the confidence intervals, Fair if 50% to 80% of the points fell within the intervals, and Poor for values less than 50%. Graphical plots of the RDB data and our measurements for the same tissue type were used to estimate the frequency-dependent edges of the confidence intervals and to count the number of measurement points within these intervals.

## 3. Results and Discussion

We measured 154 fresh human tissue samples from 14 different tissue types, obtained from surgical surplus specimens or autopsies from 30 patients. The histological analysis (described in Section 2.4) discarded six samples due to the presence of heterogeneous histological features in the punch area, or for being tumoral. Of the remaining 148 samples, 13 were categorized as necrotic tissue, 9 as fibrotic tissue, and 14 samples had fat content within the tissue. Table 2 shows the distribution of sample types measured in this study.

Figure 4 displays the measurement results for standard samples. These samples were selected following histopathological evaluation to ensure tissue homogeneity and the absence of morphological features that could interfere with permittivity measurements, such as fibrosis, necrosis, or fat infiltration—except in the case of adipose tissue, where fat is a physiological component. Only samples without such alterations were considered representative of standard tissue architecture. Where applicable, the results were compared with data from [8], and these comparisons are further elaborated in Section 3.2.

### 3.1. Detection of Fat, Necrosis, and Fibrosis Through Permittivity Measurements

The detection of fat, necrosis, and fibrosis in individual samples using complex permittivity is feasible for certain tissue types, as illustrated by the salivary gland example in Figure 5. In this case, fat content is a normal (non-pathological) condition. Higher fat content in different tissues consistently corresponded to lower permittivity values. A distinct gap is observed between the permittivity values of salivary gland samples without fat and those with fat. This gap is larger than the associated uncertainties, discussed in Section 2.5, which are primarily influenced by patient-to-patient variability (11 fat-free samples, black trace in Figure 5).

Complex permittivity measurements enabled the identification of all individual samples having fat in the liver and skeletal muscle. As shown in Figure 6 and Figure 7, both real (ε′) and imaginary (ε″) parts of the permittivity decrease in these tissues due to the abnormal fat content. Additionally, all fibrotic lung samples were detected, as fibrosis significantly increased permittivity compared to standard lung samples (Figure 8). These findings are similar to those found in [24].

The differences in the permittivity response in tissues with fat content or fibrosis may be attributed to the distinct histological characteristics of each process. Fibrosis leads to an excessive accumulation of extracellular matrix components, particularly collagen, which increases tissue density and structural stiffness. The lung, composed primarily of alveoli, contains a large amount of air, leading to low baseline permittivity. Fibrosis causes the alveoli to collapse or disappear, reducing air content and thus increasing permittivity. In contrast, tissues with fat content had a reduced permittivity probably due to the lower permittivity of lipids present within the adipocytes that infiltrate the tissue.

Necrosis detection proved to be more challenging, as it was identified only in liver samples, with lower permittivity than in both standard liver samples and those with fat content (Figure 6). No significant changes in permittivity were observed in spleen, kidney, or fat tissues with necrosis. Permittivity remained largely unchanged in these organs, despite the structural damage caused by necrosis.

Necrosis is an irreversible cellular damage resulting from membrane disruption and the release of intracellular components, which contribute to cellular and tissue degradation. The impossibility of the detection of necrosis in all individual necrotic samples in fat, spleen, and kidney tissues may not have been due to the histological process but rather to the composition of the organ (Figure 9). Fat tissue has a very low nominal permittivity value; therefore, necrosis did not significantly alter its permittivity. In fact, fat necrosis is a specific type of necrosis that releases the cytoplasm content of adipocytes into the interstitium. The cytoplasm content is composed of fat; thus, it could be a plausible explanation for the absence of changes in the measurements. The spleen, a densely packed lymphoid organ, has a naturally high nominal permittivity. The high cellular density of the organ could explain why that permittivity remained largely unaltered, even with necrosis. Similarly, the kidney also maintained relatively stable permittivity with the presence of tubular necrosis. This is probably due to its high density, being mainly composed of a complex extracellular matrix. Tubular necrosis mainly occurs in the epithelial cells of tubules immersed in this dense extracellular matrix, which could have had little influence on the measurements.

Figure 10 shows how the complex permittivity of individual abnormal samples in liver, muscle, and lung tissues compares to the nominal values of normal samples. It also illustrates how the differences in permittivity between normal and abnormal samples exceed the uncertainties, with the most pronounced differences occurring in the real part of the permittivity at low frequencies. These results highlight the potential of complex permittivity as a tool for detecting in situ certain histological features that could be integrated into the workflow of a pathology department for fresh tissue assessment.

### 3.2. Comparison with Reference Database

Figure 4 presents graphical comparisons of our measurements with those in the RDB, while Table 3 provides a detailed analysis of the data on human tissues from the RDB and our dataset for each tissue type. Overall, our data align well with the RDB down to 1 MHz, particularly for ε″. Note that the assessments in Table 3 take into account uncertainties when making comparisons with the RDB, whereas the graphs in Figure 4 do not. Our data do not align with the standard fat values reported in the RDB, so we instead compared them with RDB breast fat data (Figure 4).

**Table 3 sensors-25-02626-t003:** Summary of comparison features with human measurement data in reference database.

Type of Human Tissue	Data in RDB	Improved Freq. Density	In-Band Fit ε′ ^1^	In-Band Fit ε″ ^1^
Brain (Grey matter)	Yes	Yes	Poor	Fair
Brain (White matter)	Yes	Yes	Fair	Fair
Fat ^2^	Yes	No	Good	Good
Heart	Yes	Yes	Good	Fair
Kidney	Yes	No	Good	Good
Liver	Yes	Yes	Poor	Good
Lung	Yes	No	Poor	Good
Pancreas	No	NA	NA	NA
Salivary gland	No	NA	NA	NA
Skeletal muscle	No	NA	NA	NA
Spleen	Yes	Yes	Good	Good
Thyroid	Yes	No	Poor	Poor
Tongue	Yes	No	Poor	Fair

^1^ In-band fit with the Reference Database (RDB) was classified based on the percentage of measured permittivity values falling within the confidence intervals across the frequency range of 200 MHz to 20 GHz: Good when ≥80% of points were within the confidence intervals, Fair when 50–79%, and Poor when <50%. These intervals were determined using graphical overlays of our data and the RDB. ^2^ Comparison performed using breast fat as a reference. NA: indicates tissues not present in the RDB.

In Table 3, we can observe that, out of the 10 tissue types whose complex permittivity can be compared to the data in the RDB, 5 rate as “good” or “fair” in both ε′ and ε″ (fat, heart, kidney, spleen, and white matter), 4 rate as “good” or “fair” in ε″ and “poor” in ε′ (grey matter, liver, lung, and tongue), and 1 rates as “poor” in both ε′ and ε″ (thyroid).

A key difference between the two datasets lies in the origin and handling of the samples. Except for tongue tissues, all human samples in the RDB were obtained from autopsies performed 24 to 48 h postmortem, without histological analysis to exclude specimens affected by necrosis, fibrosis, or fat content. In contrast, the fresh human tissues used in our study were collected within 12 h postmortem or 30 min after surgical extraction, and measurements were performed at room temperature (21–24 °C), while those in the RDB were obtained at body temperature (37 °C). This temperature difference could account for part of the discrepancies between the RDB and our data, since permittivity in biological tissues is known to be temperature-dependent [25]. Furthermore, all samples in our study underwent comprehensive histological evaluation to ensure the accuracy and reliability of the permittivity measurements.

Slight modifications to our measurement system could be implemented to conduct measurements at 37 °C, enabling a fair comparison with the reference database and helping to clarify whether differences in sample handling—and the lack of histological control in the RDB—account for discrepancies in complex permittivity values. However, these modifications were not pursued, as they fall outside the scope of our current work, which is focused on assessing the potential of complex permittivity as a tool within pathology workflows.

## 4. Conclusions

We have assembled a complex permittivity measurement station to be used in a pathology department. Additionally, we have set up a rigorous protocol that ensures correspondence between permittivity measurements and histological analysis. This approach enables a consistent and reliable dataset of human tissue complex permittivity that accurately reflects histological properties, enhancing current best-practice guidelines [19,23].

Using this equipment and protocol, we conducted a comprehensive study examining the relationship between complex permittivity and histology in 14 distinct human tissue types. Our findings demonstrate significant variations in complex permittivity associated with specific histological features, whether pathological or not. Fat tissue permittivity levels are normally low, while fat content lowers permittivity in liver, skeletal muscle, and salivary gland. Necrosis reduces permittivity in liver tissue, and fibrosis substantially increases permittivity in lung tissue. These histology-dependent effects are likely the primary contributors to the discrepancies observed between our data and the RDB [8], which lacks histological control.

Despite differences in measurement protocols and the temperature at which the measurements were taken (37 °C in the RDB), our results show good or fair agreement with the RDB for both ε′ and ε″ in 5 out of 10 comparable tissue types (fat, heart, kidney, spleen, and white matter). However, the agreement is only good or fair for ε″ (and poor for ε′) in four tissue types (grey matter, liver, lung, and tongue), and poor for both ε′ and ε″ in one tissue type (thyroid).

This measurement setup, protocol, and the resulting dataset could spur future updates to the RDB [8], incorporating histological features.

Our findings further highlight the potential of permittivity-based measurements as a complementary tool in pathology workflows, particularly for the early detection of abnormal tissue conditions—such as fat infiltration, fibrosis, or necrosis—prior to FFPE processing. This approach could aid in prioritizing samples that require immediate diagnosis. Notably, while conventional pathological tissue analysis typically takes at least 12 h, permittivity measurements can be performed on fresh tissue within minutes and without the need for any processing.

Although additional studies with larger sample sizes for each tissue type are needed to achieve a more comprehensive representation, these results highlight the promise of complex permittivity as a valuable parameter for tissue characterization and diagnostic applications in pathology departments. Integrating permittivity measurements into pathology workflows could enable real-time, non-invasive assessment of tissue properties, serving as a rapid intraoperative or prescreening tool. This approach can complement conventional histological and molecular diagnostics, thereby helping to streamline and optimize diagnostic processes. Furthermore, the early detection of tissue abnormalities may support personalized treatment planning—for instance, in diseases where monitoring fibrosis progression is clinically relevant.

Nonetheless, clinical translation will require overcoming specific technical challenges, including standardization of measurement protocols, ensuring sensor stability and reproducibility, and adapting the technology for easy integration within existing pathology infrastructure. With further development, this methodology may contribute to a new generation of biomedical diagnostic strategies based on dielectric tissue properties.

## Figures and Tables

**Figure 1 sensors-25-02626-f001:**
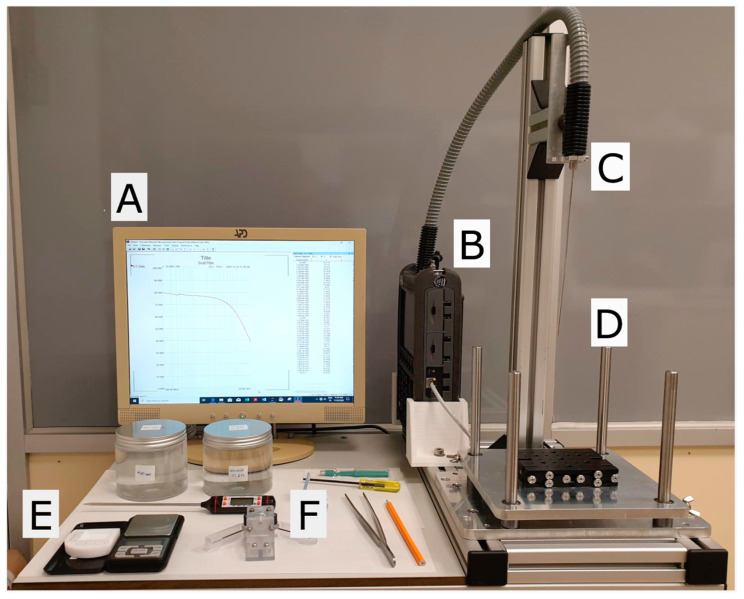
Cart used for complex permittivity measurements. It includes a PC (**A**); a VNA (**B**); a probe stand integrated in the cart’s structure (**C**); a set of coarse- and fine-lift platforms to set the sample in contact with the probe and control the probe-to-sample pressure (**D**); a PFTE sample holder and a precision scale (**E**); and forceps, thermometer, 0.1 mol/L NaCl solution, and deionized water, among other accessories for calibration and validation (**F**).

**Figure 2 sensors-25-02626-f002:**
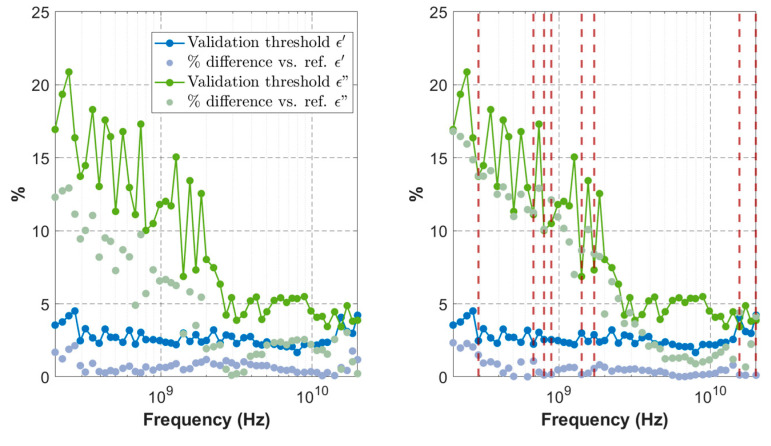
Validation Procedure: The permittivity of a 0.1 mol/L NaCl solution, measured across various frequencies, is compared with reference values from [22] (indicated by dots). Calibration is considered invalid if the deviations exceed a frequency-dependent threshold (represented by solid lines). These thresholds are established based on repeated comparisons between the measured data and the reference values. The left chart illustrates a valid calibration example, while the right chart depicts an invalid calibration, highlighting the frequencies where the discrepancies surpass the validation thresholds (dashed red lines).

**Figure 3 sensors-25-02626-f003:**
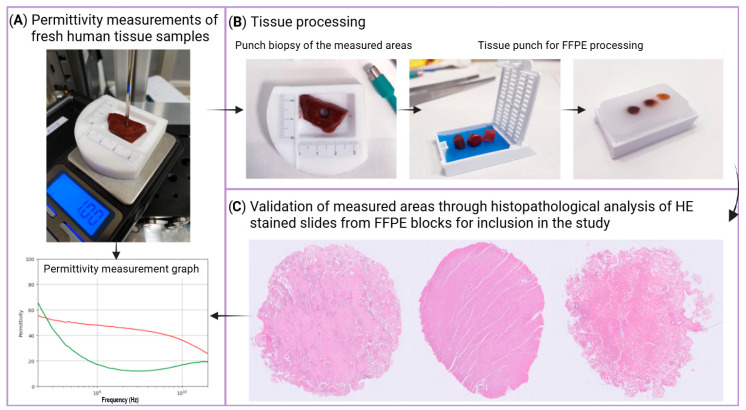
Permittivity measurements and tissue collection process: (**A**) Tissue fragment placed on the PTFE base, positioned on a scale to ensure consistent pressure from the probe, enabling accurate permittivity measurements (~1 g reading on the scale); (**B**) A punch biopsy of the measured area is performed for further formalin-fixation and paraffin-embedding (FFPE) tissue processing; (**C**) A Hematoxylin–Eosin (HE)-stained histological slide with the three different tissue types is obtained. The histopathological evaluation of the measured areas is required to ensure tissue homogeneity and validate the permittivity values for being included in the study.

**Figure 4 sensors-25-02626-f004:**
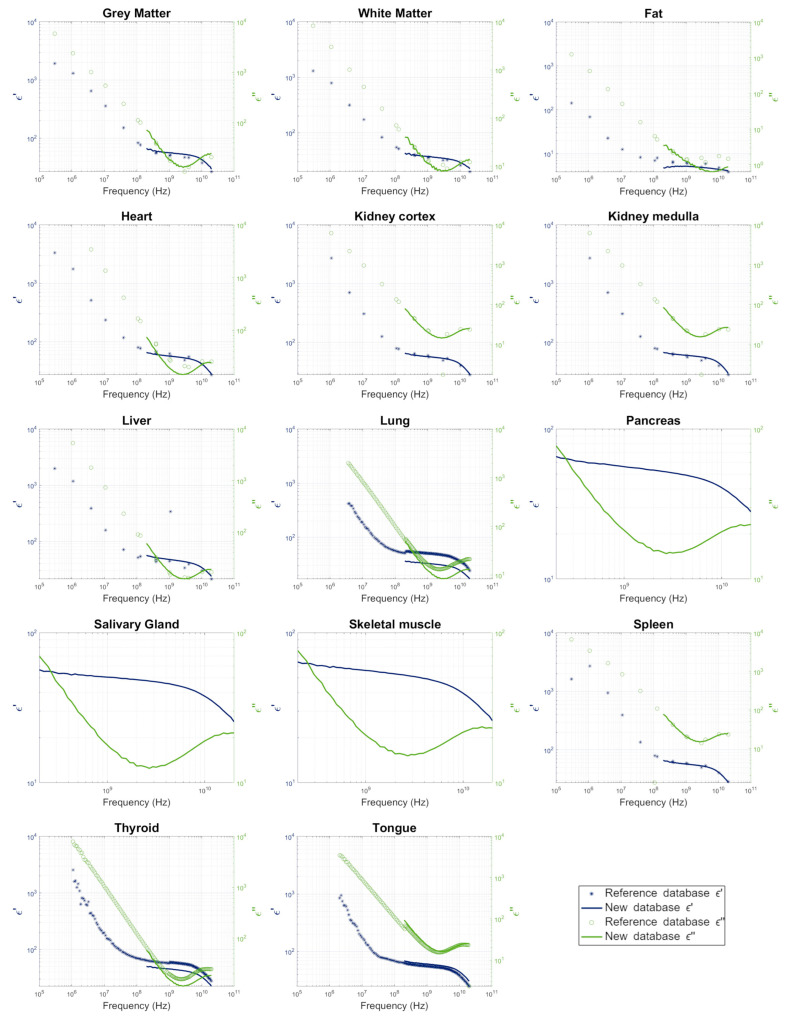
Complex permittivity of the different human tissues studied, with comparisons to the RBD data in [8], where applicable. Frequency axis (X-axis) ranges from 10^5^ to 10^11^ Hz; ε′ axis (left Y-axis) and ε″ axis (right Y-axis) range from 0 to 10^4^, with the exception of Pancreas, Salivary Gland, and Skeletal muscle, which range from 0 to 10^2^.

**Figure 5 sensors-25-02626-f005:**
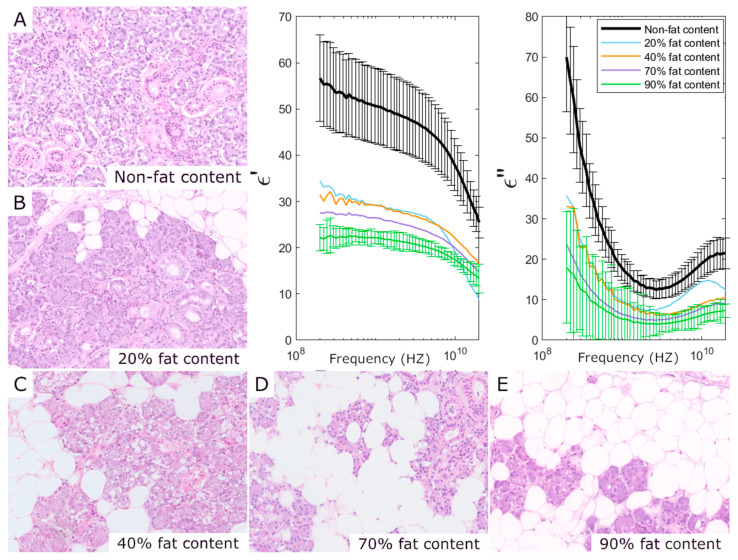
Salivary gland permittivity measurements and their corresponding histology. In the salivary gland, the presence of adipocytes is a normal condition. The relative fat content of the samples (**A**–**E**) observed in the histology can be detected through permittivity measurements. Uncertainties are derived from the standard deviation of 11 samples with no fat (in black) and instrument uncertainty for individual samples with fat (in green).

**Figure 6 sensors-25-02626-f006:**
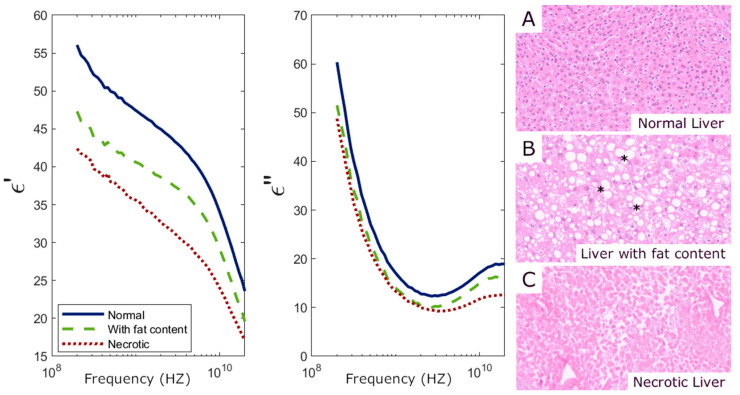
Liver permittivity measurements and their corresponding histology. (**Left**) Real (ε′) and imaginary (ε″) parts of the permittivity averaged over 12 normal tissue samples, 6 tissue samples with abnormal fat content, and data for 1 necrotic tissue sample. (**Right**) The 10× magnification Hematoxylin–Eosin-stained histological slides from the three categories represented on the permittivity graphs, i.e., (**A**) normal liver parenchyma; (**B**) liver with abnormal fat content due to steatotic liver disease, where fat vacuoles (*****) can be identified in the cytoplasm of the cells; (**C**) necrotic liver with non-viable “phantom” cells remaining.

**Figure 7 sensors-25-02626-f007:**
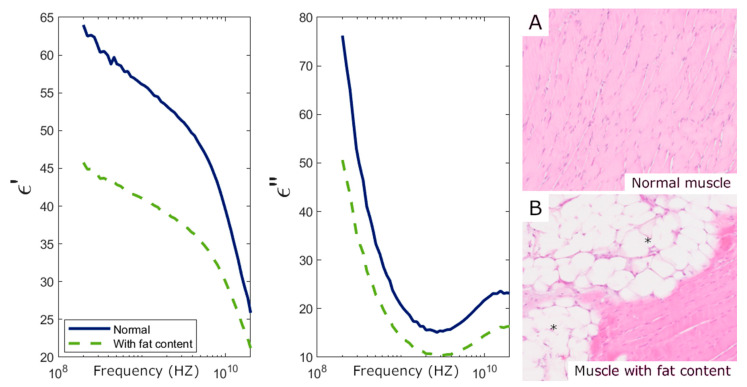
Skeletal muscle permittivity measurements and their corresponding histology. (**Left**) Real (ε′) and imaginary (ε″) parts of the permittivity averaged over 11 normal tissue samples and 4 tissue samples with abnormal fat content. (**Right**) The 10× magnification Hematoxylin–Eosin-stained histological slides from the two categories represented on the permittivity graphs, i.e., (**A**) normal skeletal muscle tissue; (**B**) skeletal muscle with abnormal fat content, where adipocytes (*****) can be identified replacing the normal muscle tissue.

**Figure 8 sensors-25-02626-f008:**
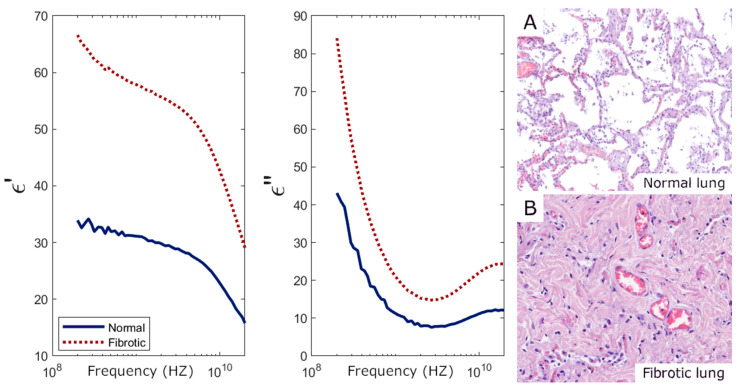
Lung permittivity measurements and their corresponding histology. (**Left**) Real (ε′) and imaginary (ε″) parts of the permittivity averaged over 14 normal tissue samples and 9 tissue samples with fibrotic areas. (**Right**) The 10× magnification Hematoxylin–Eosin-stained histological slides from the two categories represented on the permittivity graphs, i.e., (**A**) normal lung parenchyma, showing alveoli; (**B**) fibrotic lung parenchyma with no alveoli remaining.

**Figure 9 sensors-25-02626-f009:**
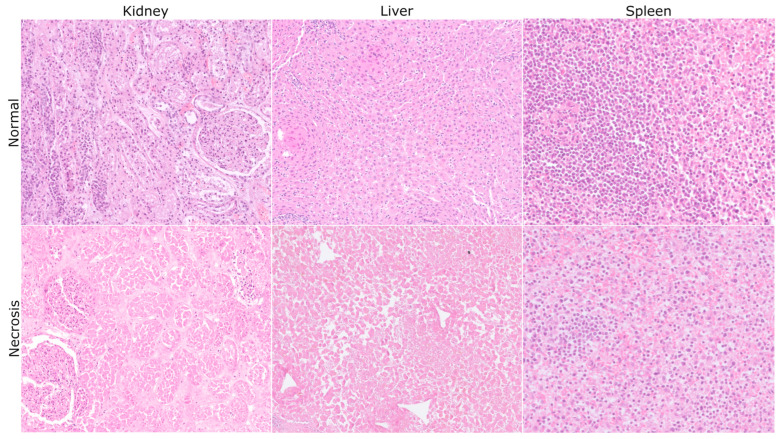
Histological images of normal (**top row**) and necrotic (**bottom row**) kidney, liver, and spleen tissues. The bottom row illustrates organ-specific structural changes associated with necrosis, which may explain the observed variability in permittivity values. The bottom-left image corresponds to a microscopic detail of a necrotic kidney that has lost its cellular detail, showing a more eosinophilic, “phantom-like” tissue containing tubules and glomeruli. The bottom-middle picture corresponds to necrotic liver parenchyma showing non-cohesive hepatocytes with a lost structure. The bottom-right picture shows a detail of spleen parenchyma with fewer lymphocytes than the normal top right picture.

**Figure 10 sensors-25-02626-f010:**
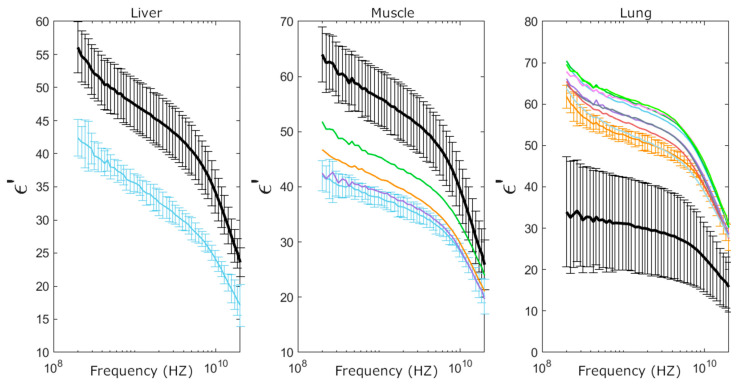
Identification of individual anomalous samples based on deviations in the real part of permittivity from nominal values for liver, skeletal muscle, and lung tissues. Black traces represent the average permittivity values of standard samples, with uncertainty margins indicating the standard deviation across multiple patients (see Table 3). Colored traces correspond to individual anomalous samples, with uncertainty margins reflecting instrumental uncertainty. Deviations observed in liver, muscle, and lung samples are attributed to necrosis, fat infiltration, and fibrosis, respectively. These graphs illustrate the feasibility of identifying anomalous samples based on permittivity measurements, despite instrumental uncertainty and the statistical variability in nominal permittivity values arising from patient-to-patient differences.

**Table 1 sensors-25-02626-t001:** Distribution of frequency points for data on human tissues in the reference database.

Type of Human Tissue	Frequency Points	Minimum Frequency (MHz)
Brain (Grey matter)	15	0.3
Brain (White matter)	15	0.3
Fat	15	0.3
Heart	15	0.3
Kidney	14	1.09
Liver	15	0.3
Lung	136	3.607
Spleen	15	0.3
Thyroid	135	1.089
Tongue	142	2.075

**Table 2 sensors-25-02626-t002:** Tissue type distribution of measured samples.

Type of Human Tissue	n	Standard	Fat Content	Necrosis	Fibrosis
Brain (Grey matter)	3	3	0	0	0
Brain (White matter)	3	3	0	0	0
Fat	17	12	0	5	0
Heart	16	16	0	0	0
Kidney (cortex)	12	9	0	3	0
Kidney (medulla)	6	4	0	2	0
Liver	19	12	6	1	0
Lung	23	14	0	0	9
Pancreas	1	1	0	0	0
Salivary gland	15	11	4	0	0
Skeletal muscle	15	11	4	0	0
Spleen	14	12	0	2	0
Thyroid	3	3	0	0	0
Tongue	1	1	0	0	0

n: total number of samples for each tissue type; Standard: number of standard samples; Fat content: number of samples with fat content (fat tissue samples are not indicated in this column, as they are classified as standard samples); Necrosis: number of samples with necrotic content; Fibrosis: number of samples with fibrotic content.

## Data Availability

The original contributions presented in this study are included in the article/Appendix A. Further inquiries can be directed to the corresponding author(s).

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
