# Peer review of "Integrating Complex Permittivity Measurements with Histological Analysis for Advanced Tissue Characterization"

_sensors, 2025, doi:10.3390/s25082626_

Round 1

Reviewer 1 Report

Comments and Suggestions for Authors

Dear authors

Very interesting article on the detection of possible diseases by measuring the permittivity of human tissue.

Could you please clarify the following points to improve the manuscript:

  • could you use your broad-spectrum measurements to identify dispersions caused by the dipole moments of macromolecules, perhaps by lowering the temperature?
  • Your calibration procedure uses a 0.1 M NaCl solution as in reference [22]. Can you specify 0.1 mol/L?
    Why this choice? Why didn't you reproduce the procedure with another concentration to check your calibration? It would be useful to add this reference  value of ε.
  •  Figure 4 is difficult to read, especially the scales and graduations. Can you improve it?
  • In table 3, in the “heart” row, the r of fair is missing (last column).
  • You say in the “comparison with reference data base” paragraph that some of the differences observed between the database and your measurements stem from the way the samples are collected and the temperature at which they are measured. Can you suggest a procedure to improve comparison?
  • You don't mention the influence of the pressure applied to the sample during measurement. Have you looked at the influence of this parameter?

    Best regards

Author Response

We would like to thank the reviewer’s insightful comments and suggestions, which have helped us to further clarify and improve various aspects of our work. We have carefully addressed each of their concerns point-by-point below

Comment 1: could you use your broad-spectrum measurements to identify dispersions caused by the dipole moments of macromolecules, perhaps by lowering the temperature?

Response: We appreciate the reviewer's suggestion. In the introduction, we reference the dependence of δ and γ dispersions on the presence of large and small molecules, respectively, to contextualize our work within the context of existing literature (see references [1] and [2]). A more detailed review of this topic can also be found in the literature review in in https://doi.org/10.1016/j.bios.2013.04.017.

Regarding the detection of macromolecules, there is published work on their characterization using dielectric measurements. For example, studies like those found in  https://doi.org/10.48550/arXiv.1806.00735 and https://doi.org/10.1021/acs.jpcb.8b02872 employ dielectric spectroscopy over a much broader frequency range than the one used in our setup. For instance, in the reference above, measurements span from 100 MHz to 2 THz (see supplemental information in https://doi.org/10.48550/arXiv.1806.00735), using a combination of three techniques for permittivity measurements and one technique for assessing conductivity. However, such approaches fall outside the scope of our current objectives, which focuses on developing a measurement setup optimized for ease of use, and repeatability, with potential integration into pathology workflows, provided that consistent and reliable correlations between permittivity data and histological features can be established.

In summary, while our system has not been evaluated for the detection of macromolecules, a brief review of the literature suggests that doing so would likely require instrumentation and methods beyond the aims and scope of our current study.

Comment 2a: Your calibration procedure uses a 0.1 M NaCl solution as in reference [22]. Can you specify 0.1 mol/L?

Response: We have updated the manuscript to specify the concentration as 0.1 mol/L, ensuring consistency with standard SI units.

Comment 2b: Why this choice?

Response: Different concentrations of sodium chloride (NaCl) solutions are commonly used to verify dielectric calibration procedures, with the selected concentration typically adjusted to approximate the permittivity of the materials being measured. In our case, 0.1 mol/L NaCl was chosen because it is widely used in dielectric measurements of biological tissues, as indicated in the measurement guidelines provided in reference [23] and the sources cited therein. A reference to [23] (Farrugia et al.) has now been included in the manuscript (line 133) at the point where the 0.1 mol/L NaCl solution is first mentioned.

Comment 2c: Why didn't you reproduce the procedure with another concentration to check your calibration? It would be useful to add this reference value of ε.

Response: The measurement guidelines provided in reference [22] (Peyman et al., 2007) recommend a single validation measurement using a sodium chloride solution of known concentration. Our experience is consistent with this recommendation. Most incorrect calibrations come from an improper calibration of the short standard or a thermal drift of the equipment, both are easily detected with a single verification solution.

There is no single permittivity value for the NaCl validation solution, since it depends on frequency, temperature, and concentration. In our study, we followed the methodology described in reference [22], which models this dependence using a Cole-Cole representation. In this model, the Cole-Cole parameters are themselves functions of both temperature and concentration, thereby capturing the full frequency-dependent behavior of the solution under varying conditions.

The manuscript has been modified in lines 132-133 to capture how the permittivity of the NaCl solution changes with frequency, temperature and concentration.

Comment 3: Figure 4 is difficult to read, especially the scales and graduations. Can you improve it?

Response: We agree that the original figure 4 could be difficult to read. To address this, we modified the figure, and now the graphs are bigger and we also added a little explanation of the scales and graduations in the figure 4 caption (lines 286-288), to facilitate the interpretation.

We thank the reviewer for this suggestion, which has improved the clarity of that figure.

Comment 4: In table 3, in the “heart” row, the r of fair is missing (last column)

Response: We have corrected it.

Comment 5: You say in the “comparison with reference data base” paragraph that some of the differences observed between the database and your measurements stem from the way the samples are collected and the temperature at which they are measured. Can you suggest a procedure to improve comparison?

Response: We thank the reviewer for this insightful suggestion. While comparison with the reference database provides a useful validation of our procedures, the primary focus of our work is not to replicate those conditions, but to establish a robust and practical measurement protocol for clinical contexts. Our long-term goal is to develop a methodology suitable for integration into pathology workflows. For this reason, all measurements were conducted in a temperature-controlled environment (21–24 °C), ensuring thermal stability across equipment, calibration standards, and tissue samples.

However, as we state in the manuscript, we recognize that the procedures we present in the manuscript could be useful to enhance databases on human permittivity. To do this, the measurement setup would have to be adapted to have the capability to bring the sample to 37ºC (the temperature used in the reference database) and control its temperature during the measurement. This could be done with a temperature-controlled measurement hotplate (instead of the PTFE holder we are currently using) and an infrared camera for thermal monitoring, although the time the tissue is at 37ºC would have to be short in order to avoid its degradation (necrosis). In all, the modified procedure would enable performing histologically controlled permittivity measurements at the same temperature as those of the reference database, but would not resolve the possible discrepancies due shortcomings in sample handling and lack of histological control of the samples used in that database.

We have added a paragraph at the end of section 3.2 (lines 422-428) to point out possible system upgrades, and why these were out of the scope of our work.

Comment 6: You don't mention the influence of the pressure applied to the sample during measurement. Have you looked at the influence of this parameter?

Response: The influence of the sample-to-probe pressure in biological samples is well known, as discussed in reference [19] and related literature. In our study, we adopted a pragmatic approach, as suggested in [19], ensuring that the applied pressure allows for consistent and repeatable measurements at the same tissue location. Specifically, we used a scale to monitor the applied pressure with a 1g scale reading and three consecutive measurements at the same point across all tissue types included in the study. This approach minimized the variability due to pressure while maintaining tissue integrity.

Reviewer 2 Report

Comments and Suggestions for Authors

This manuscript presents a comprehensive study integrating complex permittivity measurements with histological analysis of human tissues. The authors developed a measurement setup and protocol to reliably correlate complex permittivity values with specific histological features. They measured 148 fresh human tissue samples across 14 tissue types at 51 frequencies (200 MHz to 20 GHz) using an open-ended coaxial probe. The study demonstrates that complex permittivity values are strongly associated with key histological features including fat content, necrosis, and fibrosis, which could be distinguished from nominal tissue values even accounting for statistical variability and instrumental uncertainties. The work establishes a foundation for incorporating permittivity measurements into pathology workflows for fresh tissue characterization and enhancing databases with histology-controlled permittivity data.

I recommend Major/Minor Revisions for this manuscript. The paper presents valuable and novel research that is relevant to the scope of MDPI Sensors journal, but would benefit from the improvements suggested below.

Specific Comments and Questions for Authors

  1. What was the rationale for selecting the specific 51 frequencies used in this study?
  2. Page 8, Lines 275-279: The description of the normal vs. abnormal tissue samples could be clarified. Consider adding a more detailed explanation of how "standard" samples were defined.
  3. Page 11, Lines 327-334: The explanation for differences in permittivity response for tissues with fat content vs. fibrosis is insightful. Consider expanding this discussion with additional supporting references.
  4. Page 12, Lines 340-352: The discussion on why necrosis detection was challenging in certain tissues is valuable. Consider supplementing this with histological images showing the differences in necrotic tissue structure across the different organs.
  5. Page 13, Figure 9 would benefit from a clearer explanation in the caption regarding the significance of the differences observed.
  6. Page 14, Table 3: Consider explaining in more detail the criteria for "Good," "Fair," and "Poor" agreement with the reference database.
  7. The discussion of temperature differences between your measurements (21-24°C) and the reference database (37°C) should be expanded, as temperature significantly affects permittivity values.
  8. Consider discussing potential sources of biological variability in more detail (age, sex, postmortem changes, etc.).

Suggestions for Authors

  1. Elaborate on specific potential applications in pathology workflows
  2. Discuss how this approach might be integrated with other diagnostic techniques
  3. Briefly mention any technical challenges that one needs to overcome for clinical implementation
  4. Expand the statistical analysis to strengthen the findings
  5. Provide more details on the potential clinical applications of this research
  6. Discuss the temperature effects more thoroughly
  7. Consider a larger sample size for future studies, particularly for tissue types with few samples
  8. Improve Figure 3 for better clarity
  9. Add quantitative correlation between histological features (e.g., percentage of fat/fibrosis) and permittivity values

Author Response

We sincerely thank the reviewer for their detailed and valuable feedback. We are grateful for their thorough analysis of our study and the thoughtful suggestions for improvement. Their comments have greatly contributed to enhancing the clarity and depth of our manuscript. We have addressed each point raised, as outlined below.

Specific comments and Questions for Authors

Comment 1: What was the rationale for selecting the specific 51 frequencies used in this study?

Response: The frequency dependence in biological tissues is best captured on a logarithmic frequency scale, and most publications report the data in this form. In our case, our frequency scale goes from 0.2 to 20 GHz. We used an odd number of points so that frequencies in the 2-20 GHz decade are precisely ten times their counterparts in the 0.2 to 2 GHz decade. We also determined that a total of 51 points, i.e. 25 intervals per decade, is enough to capture the permittivity dependence on frequency for all the tissues under test. Using more points would have led to unnecessary long measurement times.

Comment 2: Page 8, Lines 275-279: The description of the normal vs. abnormal tissue samples could be clarified. Consider adding a more detailed explanation of how "standard" samples were defined.

Response: We appreciate Reviewer's suggestion to clarify the selection criteria for the standard samples displayed in Figure 4. In response, we have revised the manuscript to provide a more detailed explanation of these type of samples in lines 275-280.

Specifically, we selected standard tissue samples based on histopathological evaluation to ensure tissue homogeneity and the absence of morphological alterations that could interfere with the permittivity measurements, such as fibrosis, necrosis, or fat infiltration—except in the case of adipose tissue, where fat is a normal physiological component. Only samples without such alterations were considered representative of standard tissue architecture.

We hope this clarification addresses the reviewer's concern and improves the clarity of the manuscript.

Comment 3: Page 11, Lines 327-334: The explanation for differences in permittivity response for tissues with fat content vs. fibrosis is insightful. Consider expanding this discussion with additional supporting references.

Response: The frequency dependence of fat is consistently found to be below that of other tissues in many publications, among them the manuscript references [1] and [8]. Accordingly, a reduction of permittivity in tissues with fat infiltration is to be expected.

With respect to the effect of fibrosis on permittivity, in preparing this response we came up with an additional reference describing the permittivity in lung tissues (https://doi.org/10.1088/2057-1976/ad6b32), where the dielectric properties of human lung tissues—including normal, fibroelastotic (a condition characterized by fibrosis), and malignant tissues—were characterized over the frequency range of 0.5 to 10 GHz. The findings indicated that malignant lung tissue exhibited approximately 10% higher relative permittivity and conductivity compared to normal lung tissue, while fibroelastotic tissue showed dielectric properties similar to those of malignant tissue. These findings are qualitatively compatible with ours, as fibrotic samples in our study exhibited higher permittivity compared to non-fibrotic ones. However, detailed quantitative permittivity comparisons are outside of the scope of this manuscript, as they would require extended assessment of the degree and distribution of fibrosis in the samples analyzed.

An additional reference (https://doi.org/10.1088/2057-1976/ad6b32) has been cited in the discussion of effects of fibrosis in lung tissues (line 310).

Comment 4: Page 12, Lines 340-352: The discussion on why necrosis detection was challenging in certain tissues is valuable. Consider supplementing this with histological images showing the differences in necrotic tissue structure across the different organs.

Response: We thank the reviewer for its valuable feedback. We agree that supplementing the discussion on necrosis detection with histological images comparing different types of necrotic tissue across various organs would enhance the understanding of the challenges we encountered in detecting necrosis. In response to this suggestion, we have added a new figure that illustrates these tissue types (Figure 9, line 361), along with their corresponding necrotic alterations, to provide a clearer visual representation of the issue.

Comment 5: Page 13, Figure 9 would benefit from a clearer explanation in the caption regarding the significance of the differences observed.

Response: To clarify the significance of the differences observed in Figure 10 (Figure 9 before we included a new image), we have re-written the figure caption (lines 379-380, 385-387) to explicitly state that the graphs illustrate the feasibility of detecting anomalous samples through permittivity measurements, despite the presence of instrumental uncertainty and inter-patient variability.

Comment 6: Page 14, Table 3: Consider explaining in more detail the criteria for "Good," "Fair," and "Poor" agreement with the reference database.

Response: The criteria for "Good," "Fair," and "Poor" agreement with the reference database are explained in the Methods section “Comparison with a Reference Database”, lines 255-260. However, in response to the reviewer's comment, we have revised the explanation provided in Table 3 (lines 399-404) to ensure consistency with the description in the Methods section. We hope this revision clarifies the criteria and improves the readability of the table.

Comment 7: The discussion of temperature differences between your measurements (21-24°C) and the reference database (37°C) should be expanded, as temperature significantly affects permittivity values.

Response: The temperature dependence could be a possible reason for disagreement between our results and those of the reference database. Following both reviewers’ valuable suggestions, we have included an explanation and a reference that describes such dependence in lines 417 - 419.

Comment 8: Consider discussing potential sources of biological variability in more detail (age, sex, postmortem changes, etc.).

Response: We agree that this variability sources should be analyzed, and we will consider this in future upgrades of this work. At this time, we feel this is not viable due to time limitations to prepare a new version of the manuscript, and the small number of samples for most tissue types, which would require a new measurement campaign.

Suggestions for Authors

Comments 1, 2, 3 and 5: Elaborate on specific potential applications in pathology workflows; Discuss how this approach might be integrated with other diagnostic techniques; Briefly mention any technical challenges that one needs to overcome for clinical implementation; Provide more details on the potential clinical applications of this research.

Response: We thank the reviewer for these suggestions. In response, we have expanded the conclusion section, lines 456 to 479, to provide a more comprehensive discussion of the potential clinical applications of permittivity-based measurements in pathology workflows. We elaborate on how this approach could be integrated with other diagnostic techniques, offering real-time, non-invasive assessments of tissue properties that could serve as rapid, intraoperative or prescreening modalities. This could provide valuable support to conventional histological or molecular diagnostics. Additionally, we briefly address the technical challenges to clinical implementation, such as the standardization of measurement protocols, sensor stability, and integration with existing pathology infrastructure. We also highlight how early detection of abnormalities, such as fibrosis, could significantly improve personalized treatment planning and patient outcomes. We hope these additions clarify the potential impact and practical applications of our research.

Comment 4: Expand the statistical analysis to strengthen the findings

Response: Due to the limited sample size, we are unable to conduct a statistically significant analysis at this stage. A larger sample set would be necessary for robust statistical comparisons, and we plan to address this in future studies.

Comment 6: Discuss the temperature effects more thoroughly

Response: Temperature effects are difficult to describe. A thorough description is given in doi:10.3390/s19071707 where the temperature dependence of the Cole-Cole parameters for several tissues is fitted.  The real part of the permittivity tends to decrease with increasing temperature, and the conductivity part has a cross-over behavior (decreases for frequencies higher than about 3 GHz and increases for frequencies below that).  This dependence is had to capture and discuss in the manuscript without shifting its focus away from its main line.  To avoid shifting the focus of our discussion, we have mentioned this temperature dependence and included this publication in our reference list (reference number 25, citated in the manuscript in line 419)

Comment 7: Consider a larger sample size for future studies, particularly for tissue types with few samples

Response: We thank the reviewer for the suggestion. We fully agree with the need to expand sample size in the future. We consider this work as a first step to explore viability.

Comment 8: Improve Figure 3 for better clarity

Response: We agree on the need to improve this figure. We have updated the figure to improve clarity and readability (line 190).

Comment 9: Add quantitative correlation between histological features (e.g., percentage of fat/fibrosis) and permittivity values

Response: We thank the reviewer for this valuable suggestion. We fully agree with the importance of establishing a quantitative correlation between histological features and permittivity values. However, we believe that a larger sample size is required to accurately determine such a correlation and establish reliable thresholds. It is also important to consider the uncertainties inherent in the setup, which could impact the precision of these measurements. Future studies with more extensive datasets will be necessary to address this and enable a more robust quantitative analysis.